# Effects of neuromuscular training on knee proprioception in individuals with anterior cruciate ligament injury: a systematic review and GRADE evidence synthesis

Ashokan Arumugam ,[1] Martin Björklund,[2,3] Sanna Mikko,[2] Charlotte K Häger[2]

► Prepublication history and additional online supplemental material for this paper are available online. To view these files, please visit the journal online (http://dx.doi.org/10.1136/bmjopen-2021-049226).

¹Department of Physiotherapy, College of Health Sciences, University of Sharjah, Sharjah, UAE
²Department of Community Medicine and Rehabilitation – Physiotherapy Section, Umeå University, Umeå, Sweden
³Centre for Musculoskeletal Research, Department of Occupational Health Sciences and Psychology, Faculty of Health and Occupational Studies, University of Gävle, Gävle, Sweden

**Correspondence to**
Professor Charlotte K Häger; charlotte.hager@umu.se

## ABSTRACT

**Objective** To systematically review and summarise the evidence for the effects of neuromuscular training compared with any other therapy (conventional training/sham) on knee proprioception following anterior cruciate ligament (ACL) injury.

**Design** Systematic Review.

**Data sources** PubMed, CINAHL, SPORTDiscus, AMED, Scopus and Physical Education Index were searched from inception to February 2020.

**Eligibility criteria** Randomised controlled trials (RCTs) and controlled clinical trials investigating the effects of neuromuscular training on knee-specific proprioception tests following a unilateral ACL injury were included.

**Data extraction and synthesis** Two reviewers independently screened and extracted data and assessed risk of bias of the eligible studies using the Cochrane risk of bias 2 tool. Overall certainty in evidence was determined using the Grading of Recommendations, Assessment, Development and Evaluation (GRADE) tool.

**Results** Of 2706 articles retrieved, only 9 RCTs, comprising 327 individuals with an ACL reconstruction (ACLR), met the inclusion criteria. Neuromuscular training interventions varied across studies: whole body vibration therapy, Nintendo-Wii-Fit training, balance training, sport-specific exercises, backward walking, etc. Outcome measures included joint position sense (JPS; n=7), thresholds to detect passive motion (TTDPM; n=3) or quadriceps force control (QFC; n=1). Overall, between-group mean differences indicated inconsistent findings with an increase or decrease of errors associated with JPS by ≤2°, TTDPM by ≤1.5° and QFC by ≤6 Nm in the ACLR knee following neuromuscular training. Owing to serious concerns with three or more GRADE domains (risk of bias, inconsistency, indirectness or imprecision associated with the findings) for each outcome of interest across studies, the certainty of evidence was very low.

**Conclusions** The heterogeneity of interventions, methodological limitations, inconsistency of effects (on JPS/TTDPM/QFC) preclude recommendation of one optimal neuromuscular training intervention for improving proprioception following ACL injury in clinical practice. There is a need for methodologically robust RCTs with homogenous populations with ACL injury (managed conservatively or with reconstruction), novel/well-designed neuromuscular training

### Strengths and limitations of this study

► A systematic review of neuromuscular training on knee proprioception following the Preferred Reporting Items for Systematic Reviews and Meta-Analyses guidelines, using a broad search in six electronic databases.
► The risk of bias associated with the outcomes of interest (knee proprioception measures) in the included RCTs was assessed using the updated Cochrane risk of bias 2 tool.
► The overall certainty of evidence for the effects of neuromuscular training on knee joint position sense, threshold to detect passive motion, and quadriceps force control following ACL injury/reconstruction was ascertained using the Grading of Recommendations, Assessment, Development and Evaluation tool.
► Only RCTs published in English were included.
► A meta-analysis was precluded because of clinical heterogeneity of interventions and outcome measures.

and valid proprioception assessments, which also seem to be lacking.
**PROSPERO registration number** CRD42018107349.

## INTRODUCTION

Anterior cruciate ligament (ACL) injury is a common musculoskeletal injury[1 2] accounting for an annual incidence rate of 68.6/100 000 person-years in the USA.[3] ACL injury is most prevalent in young athletes (14–18 years for females and 19–25 years for males).[3] The injury occurs more often during competition rather than training, with ~70% or more of the injuries representing noncontact mechanisms[4 5] such as landing from a jump, sudden deceleration and/or while cutting.[6] Thus, the injury mechanisms are related to neuromotor control, among other factors, of the individual. ACL injury is predominantly treated



by surgical reconstruction,[3] and followed by a long period of rehabilitation and yet many individuals do not return to preinjury levels of activity[7] which challenges the efficacy of existing preventative and rehabilitative strategies.

Individuals with an ACL injury present with a decreased number of proprioceptive mechanoreceptors (Pacinian capsules, Ruffini nerve endings and Golgi tendon organs),[8 9] which might alter somatosensory input to the central nervous system (CNS)[9] leading to decreased knee proprioception. Disturbed proprioception might also be caused by acute inflammation and pain, and the capsule and surrounding ligaments getting affected following instability.[10 11] Although there has been a debate regarding the effects of ACL injury on different knee proprioception tests,[2 12] our recent systematic review[13] suggests that knee joint position sense (JPS tests have sufficient validity in discriminating ACL-injured knees from asymptomatic knees (accepted). When compared with non-injured controls, individuals with ACL injury demonstrate altered movement strategies,[4 14] quadriceps muscle weakness[15] and onset and progression of osteoarthrosis.[6 16] Due to the potential serious consequences of the injury, much attention and clinical efforts have been dedicated to preventative and rehabilitative strategies for ACL injury,[11] including various neuromuscular training (NT) methods believed to improve the proprioceptive ability.

Even if proprioceptive deficits could affect neuromotor control, the rationale, mechanisms and plausibility for improving proprioception by training need to be verified. In the context of neuroplasticity, functional MRI has revealed that individuals with ACL-deficient knees demonstrate less activation in several sensorimotor cortical areas and increased activation in presupplementary motor areas, posterior secondary somatosensory area and posterior inferior temporal gyrus compared with controls with asymptomatic knees during a knee flexion-extension task.[1] It seems individuals with ACL reconstruction (ACLR) adapt a visual-sensory-motor strategy instead of a normal sensorimotor strategy owing to aberrant sensory feedback following ACL injury.[17] Nevertheless, neuroplastic reorganisation ensues where other potential sensory sources are used to organise the movement or regulate neuromotor control, particularly in (sporting) tasks with higher complexity. Therefore, ACL injuries might be regarded as a neuromotor control dysfunction rather than a simple peripheral musculoskeletal injury.[11 18] It is yet unclear though whether NT can improve proprioception after an ACL injury[11 19] and the neurophysiological mechanisms underpinning such interventions need further substantiation.

To date, there is no consensus on the most effective rehabilitation programmes for ACL injury, and the prevalence of reinjury after returning to sport is up to 30%.[18] Owing to the neuroplastic changes and possibly altered proprioception following an ACL injury, NT has received much attention to enhance dynamic joint stability and relearn movement patterns and skills.[20] In this context, both NT and sensorimotor training terms have been used. NT is defined as '…training enhancing unconscious motor responses by stimulating both afferent signals and central mechanisms responsible for dynamic joint control'[20] and sensorimotor training aims to improve '…function of the CNS in regulating movement in order to reach proper firing patterns for maintaining joint stability…'.[21] Active knee motion will in any case stimulate proprioceptors, which in turn would alter the demands on the CNS.[10 19] Henceforth, we will use the term NT in this review.

There are different ways to challenge proprioception, for example, vibration may be used to alter afferent input from muscle spindles; an unstable surface can challenge input from the ankle; vision can be occluded or head position can be changed to disturb visual and vestibular information,[10] or focus can be shifted to influence cognitive processing sources.[18] Due to a putative visual sensory motor strategy following ACL injury, a modified visual feedback training method might decrease visual reliance and improve sensorimotor function.[18] Most studies exploring the effects of NT on proprioception combine different exercises and various outcome measures which precludes isolating the effects of a proprioception-specific exercise.[22] Therefore, this study aimed to systematically review and summarise the evidence for the effects of NT compared with comparator/control interventions on proprioception measured by knee-specific proprioception tests in individuals with ACL injury or reconstruction.

## METHODS

We adhered to the Preferred Reporting Items for Systematic Review sand Meta-Analyses (PRISMA) checklist[23] and the reporting guidelines for Synthesis Without Meta-analysis in systematic reviews.[24] A list of acronyms used in the review is summarised in table 1.

### Eligibility criteria

The structure of PICOS[25] was used to frame the following criteria:

1. Participants: Individuals aged over 15 years (both sexes) with a history of a unilateral ACL rupture, managed conservatively or surgically reconstructed, with or without concomitant meniscus and/or collateral ligament injuries on the injured leg, without any other lower extremity injuries/surgeries that would confound the outcomes of rehabilitation training.
2. Intervention: Specific NT, closed or open kinetic chain exercises, balance training, joint repositioning training, joint force sense training, coordination training, plyometric training, whole-body vibration, virtual gaming training, an accelerated rehabilitation protocol or any other training programmes focusing on improving the lower limb neuromuscular control and knee proprioception.

**Table 1** A list of acronyms used in the review

| Acronym | Definition |
| --- | --- |
| ACL | Anterior cruciate ligament |
| ACLR | Anterior cruciate ligament reconstruction |
| AAE | Absolute angular error |
| CNS | Central nervous system |
| GRADE | Grading of recommendations, assessment, development and evaluation |
| JPS | Joint position sense |
| NT | Neuromuscular training |
| PRISMA | Preferred reporting items for systematic review and meta-analysis |
| PICOS | Participants, intervention, comparator, outcome measures, study design |
| QFC | Quadriceps force control |
| RCT | Randomised controlled trial |
| ROB | Risk of bias |
| TTDPM | Thresholds to detect passive motion |
| WBVT | Whole-body vibration therapy |

3. Comparator: Any other therapy, conventional training, usual care, placebo or sham therapy.
4. Outcome measures: Knee-specific proprioception tests targeting JPS, kinesthesia (threshold to detect passive motion (TTDPM)), force sense/perception, active movement extent discrimination, velocity sense or psychophysical threshold methods[13]; they can be performed actively and/or passively with or without visual input in weight bearing or non-weight bearing positions.[10]
5. Study design: randomised controlled trials (RCTs) or controlled clinical trials.

## Data sources and searches

Database-specific search terms (eg, Medical Subject Headings (MeSH)) were combined using Boolean operators ('AND' and 'OR') under three conceptual domains: participants, interventions and outcomes. Six electronic databases were searched from their inception to 12 February 2020: PubMed, Cumulative Index to Nursing & Allied Health Literature (CINAHL via EBSCOhost), SPORTDiscus (via EBSCOhost), the Allied and Complementary Medicine Database (AMED via EBSCOhost), Scopus and Physical Education Index (via Proquest) (online supplemental file 1).

## Study selection

One reviewer (SM) imported all titles and abstracts retrieved from the databases into EndNote X8. Two reviewers (AA and SM) independently checked titles, abstracts and/or full text by following a screening questionnaire (online supplemental file 2). Any disagreements in inclusion of articles were adjudicated by two other reviewers (CKH and MB) until consensus was reached. A manual search of the reference lists of included articles was performed.

## Data extraction

Data were extracted by one reviewer (SM) and verified by another reviewer (AA) using a customised data extraction sheet (online supplemental file 3). If any data were missing, the corresponding authors were contacted via email.

## Quality assessment

The risk of bias (ROB) for each outcome of interest in the included studies was evaluated using the Cochrane ROB 2 tool.[26] The tool has five domains: (1) randomisation (number of signalling questions (n=3), (2) deviations from intended interventions (n=7), (3) missing outcome data (n=5), (4) measurement of the outcomes (n=5) and (5) selection of the reported results (n=3). Each signalling question can be answered as (1) yes, (2) probably yes, (3) probably no, (4) no and (5) no information. Responses to the questions provide the basis for judgement of the ROB at each domain level using a tool-specific algorithm resulting in one out of three possible judgements: (1) low ROB, (2) some concerns or (3) high ROB. An overall ROB score for each outcome in a study can be low (with a low ROB for all domains), some concerns (if some concerns prevail in at least one domain without a high ROB for any domain) or high (if a high ROB underpins at least one domain or some concerns remain in multiple domains, defining multiple as more than two).

## Evidence synthesis

The overall evidence level in this review was determined using the Grading of Recommendations, Assessment, Development and Evaluation (GRADE) tool considering the following five domains: (1) ROB: high risk, some concerns or low risk associated with knee proprioception measures based on the Cochrane ROB 2 tool; (2) inconsistency of findings: similar or conflicting direction of effect, effect estimates and overlap of confidence intervals for knee proprioception measures from different studies; (3) indirectness of evidence: appropriateness of participants, interventions and outcomes used to answer the review question; (4) imprecision of results: the length of 95% confidence intervals (CIs) of effect estimates and overall sample (number of participants) from which effect estimates are derived; and (5) other domains: for example, publication bias if applicable.[27] The overall evidence was rated as very low, low, moderate or high.

A meta-analysis was precluded owing to clinical heterogeneity of interventions and outcome measurements (JPS, TTDPM and quadriceps force control (QFC)). For instance, despite seven studies targeting JPS, a meta-analysis was not appropriate because at most two studies used the same method (active-active,[28 29] passive-passive[30 31] or passive-active)[32 33] but the starting and target angles and the number of trials per each



angle varied between these proprioception tests in the included studies. Further, the neuromuscular training interventions, targeting JPS, widely varied between studies[28–34]: closed kinetic chain exercises on a balance pad,[34] whole-body vibration therapy (WBVT),[29 30] motor control exercises for the lower limbs,[32] backward walking on a treadmill,[31] Nintendo Wii Fit training[28] and cross-education of strength training of the non-injured leg along with standard rehabilitation.[33] Further, in addition to inconsistent findings among the studies, a significant statistical heterogeneity ($I^2 > 60\%$) in a random-effects meta-analysis was evident. Although meta-analyses were excluded, the Review Manager V.5.3 software (the Cochrane Collaboration) was used to calculate between-group mean differences (effect sizes) and their 95% CIs for summarising the findings for each outcome of interest in table 2.

### Patient and public involvement
Neither patients nor public were involved.

## RESULTS
### Search results
Electronic databases search led to a total of 2706 articles (excluding duplicates: 2162). After title and abstract screening, 22 articles were shortlisted for full-text screening and subsequently nine articles met the inclusion criteria (figure 1). Thirteen articles were excluded owing to the following reasons: not an RCT (n=1),[35] no knee-specific proprioception tests (n=6),[36–41] participants were without an ACL injury (n=1),[42] knee proprioception data were missing and the corresponding author did not respond to our emails (n=1),[43] a comparison between different surgical intervention groups with same rehabilitation programme (n=2),[44 45] and lack of a neuromuscular rehabilitation training programme (n=2).[46 47] No additional relevant studies were identified through manual search of bibliographic references.

### Study design and participants
All the nine studies included were RCTs with a total of 386 participants and two studies had their trial preregistered in a clinical trial registry.[31 33] All participants had undergone an ACLR with a bone-patellar-tendon-bone or a hamstring graft (table 2).

### Quality assessment
The agreement (Cohen's kappa) of responses to the signalling questions between the two reviewers (AA and MB) was substantial (0.69±0.047, p<0.001). Disagreements were discussed and resolved by the two reviewers. Online supplemental figure 1 shows the percentage of studies judged as low risk, some concerns and high ROB in the five domains, and table 3 shows domain judgements of each study. The overall ROB judgement showed that four of the included studies had a high

ROB,[28 29 32 34] four had some concerns,[30 31 48 49] and one study[33] had a high ROB for JPS and some concerns for QFC. The domain that most consistently showed ROB across studies was bias in selection of the reported results (online supplemental figure 1 and table 3). The most common reason was the absence of information regarding prespecified plan of analyses. None of the included studies reported trial protocol publication and only two[31 33] reported trial registration. Furthermore, two studies were judged to perform inappropriate multiple analyses.[28 29] Judgement of bias in measurement of the outcome (domain 4, table 3) showed most scattered results across studies (online supplemental figure 1). A high ROB was found in three studies of which one had no information on measurements[34] and two showed inappropriate measurement methods of the outcome of interest.[28 33] In the study by Zult et al, only one trial per target was performed to estimate JPS,[33] while Baltaci et al used a test with presumably a high demand on motor and memory components,[28] without reporting its reliability or validity. The domain with least ROB was missing outcome data where all studies, except one,[32] had low ROB.

### Rehabilitation programmes
The studies included a spectrum of rehabilitation programmes employed to influence knee proprioception (table 2). Only one study by Baltaci et al investigated the effects of using feedback with an external focus in a simulated sport-specific gaming environment with Nintendo Wii Fit compared with conventional rehabilitation.[28] On the contrary, the remaining eight studies focused on having an internal focus (mainly related to the position of specific body parts) for NT. Two studies[29 30] explored the effects of WBVT combined with or without conventional rehabilitation compared with conventional rehabilitation alone. Cho et al compared closed kinetic chain exercises on a balance pad versus on a stable floor.[34] Risberg et al compared the effects of an NT compared with strength training. In their neuromuscular programme, the first half of the rehabilitation focused on exercises on a wobble board or trampoline and exercises to increase the range of motion, while the end of the programme focused on specific training of plyometric, agility and sport-specific skills.[48] Beynnon et al evaluated the effects of accelerated (19 weeks) vs non-accelerated (32 weeks) programmes of conventional training.[49] The timeframe and exercises in their experimental programme ranged from 1 to 7 weeks for range of motion and muscle activation, 8–11 weeks for dynamic functional activities such as biking and jogging, and finally, 12–19 weeks for plyometric and agility drill exercises.[49] Kaya et al studied the effects of neuromuscular (motor control) exercises for the lower limbs combined with standard rehabilitation compared with standard rehabilitation alone.[32] Shen et al examined the outcome of standard rehabilitation combined with backward walking at 1.3 km/hour on a treadmill

**Table 2** Summary of study characteristics

| Study citation | Sample size*, age (mean±SD), gender; ACLR (Graft) | Intervention; adherence to prescribed exercises/training | Comparator; adherence to prescribed exercises/training | Knee-specific proprioception test; outcome | Between-group (experimental vs control) comparisons of ACL-injured (reconstructed) limb - mean difference (95% CI)† |
|---|---|---|---|---|---|
| Baltaci et al (2013)[28] | Exp: n=15, 28.6±6.8 years, 15 men; Com: n=15, 29.3±5.7 years, 15 men; ACLR (hamstring tendon graft). | Nintendo Wii Fit training: three times/week; 60 min/session; from week 1–12 after ACLR. **Adherence:** NR | Conventional rehabilitation: Week 1–12 after ACLR; **Adherence:** NR | **Proprioception test:** JPS (ipsilateral replication method); **Body position:** NR; **Instrument:** Monitored Rehab System with a leg press machine and a computer game; **Procedure:** Active-active, with and without blindfolding of the eyes (two trials each); **Starting angle (SA):** NR; **Target angle (TA):** NR; **Outcome measure:** absolute angular error (AAE; difference between visual and non-visual results for each leg) | **JPS‡ at 12 weeks postintervention:** 1.90(−31.20 to 35.00) 33.30(−28.02 to 94.62) |
| Beynnon et al (2011)[49] | Int: n=19, 29.7±10.1 years, 13 males, 6 females; Com: n=17, 30.2±9.9 years, nine males, 8 females; ACLR (patellar tendon graft) | Accelerated rehabilitation: daily exercises at home +3 times/week exercises under supervision from week 1–19 after ACLR; **Adherence:** 94% (range, 25%–292%) over 19 weeks | Non-accelerated rehabilitation: daily exercises at home +3 times/week exercises under supervision from Week 1–32 after ACLR; **Adherence:** 53% (range, 13%–108%) over 32 weeks | **Proprioception test:** TTDPM; **Body position:** Seated; **Instrument:** A customised joint motion detection system; **Procedure:** passive movement of the knee into flexion or extension (three trials for both ACL-reconstructed and contralateral uninjured knees) with eyes blindfolded; **SA:** NR; **Angular velocity:** 0.1°/s; **Outcome measure:** Threshold angle (difference between the initial angle (SA) and the angle at which the test was stopped) to detect passive knee motion into flexion or extension (mean of the three trials in one direction). | **TTDPM (°)‡at 24 months post-ACLR: SA (NR):** 0.09(−0.42 to 0.60) |
| Cho et al (2013)[34] | Int: n=14, 29.92±5.46 years; 14 males; Com: n=14, 28.78±7.24 years; 14 males; ACLR (NR). | Unstable exercise group: exercises performed on a balance pad or balance board; 60 min/ session; three times/week early after injury, for 6 weeks; **Adherence:** NR | Stable exercise group: exercises performed on a stable floor: 3 times/week Early after injury, for 6 weeks; **Adherence:** NR | **Proprioception test:** JPS; **Body position:** seated (?); **Instrument:** Biodex dynamometer; **Procedure:** NR-active, with eyes blindfolded; **SA:** 90°; **TA:** 15°, 45°; **Outcome measure:** AAE (mean of the three trials at each angle). | **JPS (°)** at 6 weeks post intervention:** TA 15°: 0.14(−0.69 to 0.97) TA 45°: −0.87(−1.91 to 0.17) |

**Table 2** Continued

| Study citation | Sample size*, age (mean±SD), gender; ACLR (Graft) | Intervention; adherence to prescribed exercises/training | Comparator; adherence to prescribed exercises/training | Knee-specific proprioception test; outcome | Between-group (experimental vs control) comparisons of ACL-injured (reconstructed) limb - mean difference (95% CI)† |
|---|---|---|---|---|---|
| Fu et al (2013)[30] | Int: n=24, 23.3±5.2 years; Com: n=24, 25.2±7.3 years; ACLR (hamstring graft). | Conventional rehabilitation program +Whole body vibration therapy: 2 times/week from week 5–13 after ACLR; **Adherence:** 83.2% over 12 weeks | Conventional rehabilitation programme: week 5–13 after ACLR; **Adherence:** 84.4% over 12 weeks | **Proprioception test:** JPS; **Body position:** seated; **Instrument:** Biodex dynamometer; **Procedure:** passive-passive, eyes blindfolded; **SA:** 90°; **TA:** 30°, 60°; **Outcome measure:** AAE (mean of the three trials at each angle) | **JPS (°)‡ at 6 months post-ACLR:** **TA 30°:** −0.82(−2.69 to 1.05) **TA 60°:** −0.70(−2.31 to 0.91) |
| Kaya et al (2019)[32] | Int (Group 1): n=20; 29.35±9.71 years; 20 males; Com (Group 2): n=20; 31.60±8.45 years; 20 males; ACLR (tibialis anterior allograft). | Standard rehabilitation programme (0–2 weeks)+neuromuscular control exercises (3–36 weeks); **Adherence:** NR | Standard rehabilitation programme (0–36 weeks); **Adherence:** NR | **Proprioception test:** JPS; **Body position:** seated (?); **Instrument:** Biodex dynamometer; **Procedure:** passive-active, eyes blindfolded; **SA:** 90°; **TA:** 15°, 45°, 75°; **Outcome measure:** AAE (mean of six trials at each angle) | **JPS (°)‡ at 24 months post-ACLR:** **TA 15°:** −1.51(−3.30 to 0.28) **TA 45°:** −1.69(−5.06 to 1.68) **TA 75°:** −1.30(−3.34 to 0.74) |
| Moezy et al (2008)[29] | Int: n=12, 24.51±3.38 years; Com: n=11, 22.70±3.77 years; ACLR (patellar tendon graft) | Whole-body vibration therapy: 3 times/week from week 12–16 after ACLR; **Adherence:** NR | Conventional strengthening exercises programme: 3 sessions/week Week 12–16 after ACLR; **Adherence:** NR | **Proprioception test:** JPS; **Body position:** seated; **Instrument:** Biodex dynamometer; **Procedure:** active-active, eyes blindfolded; **SA:** 90°; **TA:** 30°, 60°; **Outcome measure:** AAE (mean of five trials at each angle for both ACL-reconstructed and contralateral uninjured knees) | **JPS (°)*at 4 months post-ACLR:** **TA 30°:** 1.66(−0.40 to 3.72) **TA 60°:** 3.03(1.54 to 4.52) |

Continued

**Table 2** Continued

| Study citation | Sample size*, age (mean±SD), gender; ACLR (Graft) | Intervention; adherence to prescribed exercises/training | Comparator; adherence to prescribed exercises/training | Knee-specific proprioception test; outcome | Between-group (experimental vs control) comparisons of ACL-injured (reconstructed) limb - mean difference (95% CI)† |
|---|---|---|---|---|---|
| Risberg et al (2007)[48] | Int: n=39; three females - 27.2 (range: 20.6–37.9) years and 26 males - 27.7 (16.7–39.6) years; Com: n=35, 14 females - 26.5 (19.8–38.0) years and 21 males - 31.2 (19.4–40.3) years; ACLR (patellar tendon graft) | Neuromuscular training programme: 2–3 times/week from week 1–24 after ACLR; **Adherence:** 71% over ~20 weeks | Traditional strength training: 2–3 times/week from week 1–24 after ACLR; **Adherence:** 91% over ~20 weeks | **Proprioception test:** TTDPM; **Body position:** NR; **Instrument:** a customised TTDPM device; **Procedure:** passive movement of the knee into flexion and extension (three trials for each direction for both ACL-injured knees and contralateral uninjured knees); no information on blindfolding of eyes; **SA:** 15°; **Angular velocity:** 0.5°/s; **Outcome measure:** Threshold angle (difference between the SA and the angle at which the test was stopped) to detect passive knee motion into flexion or extension mean of the three trials in one direction (mean of three trials for each angle in each direction). | **TTDPM (°)‡ at 6 months post-ACLR:** **SA 15°:** −0.02(−0.39 to 0.35) (note: TTDPM data were available only for the first 47 participants out of 74 in total). |

Continued

**Table 2** Continued

| Study citation | Sample size*, age (mean±SD), gender; ACLR (Graft) | Intervention; adherence to prescribed exercises/training | Comparator; adherence to prescribed exercises/training | Knee-specific proprioception test; outcome | Between-group (experimental vs control) comparisons of ACL-injured (reconstructed) limb - mean difference (95% CI)† |
|---|---|---|---|---|---|
| Shen *et al* (2019)[31] | Int (A): n=10; 36.6±12.1 years; five male, 5 females. Int (B): n=11; 37.5±9.39 years; six male, 5 females. Int (C): n=11; 34±10.29 years; seven male, 4 females. Int (D): (n=10); 32.9±11.45 years; six male, 4 females. Com: n=10; 35.5±10.1 years; seven male, 3 females; ACLR (patellar tendon graft, hamstring tendon graft, allograft) | Standard rehabilitation +backward walking on the treadmill: Int. groups A, B, C, and D underwent backward walking training at 1.3 km/h at different inclination angles of the treadmill (0°, 5°, 10°, and 15°, respectively); 20 min/day, 5 days/week for 4 weeks; **Adherence:** NR | Standard rehabilitation with range of motion exercises, power exercises, walking, and cycling (duration and other parameters: NR); **Adherence:** NR | **Proprioception test 1:** JPS; **Body position:** supine lying; **Instrument:** continuous passive motion device; **Procedure:** passive-passive, eyes blindfolded; **SA:** 0°; **TA:** 20°, 50°, 80°; **Outcome measure:** AAE (mean of the three trials at each angle for ACL-injured knees?). **Proprioception test 2:** TTDPM; **Body position:** Supine lying; **Instrument:** continuous passive motion device; **Procedure:** passive movement of the knee into flexion (three times for each angle for ACL-injured knees?) with eyes blindfolded; **SA:** 20°, 50°, 80°; **Angular velocity:** 1°/s; **Outcome measure:** Threshold angle to detect passive knee motion into flexion (mean of three trials for each angle in one direction). | Int (A) vs Com group at 1-month postintervention§: **JPS (°)‡:** **TA 20°:** −1.40(−2.59 to −0.21) **TA 50°:** −1.36(−2.35 to −0.37) **TA 80°:** −1.28(−2.31 to −0.25) **TTDPM (°)‡: SA 20°:** −1.34(−2.11 to −0.57) **SA 50°:** −1.40(−2.05 to −0.75) **SA 80°:** −1.29(−2.00 to −0.58) |

Continued

**Table 2** Continued

| Study citation | Sample size*, age (mean±SD), gender; ACLR (Graft) | Intervention; adherence to prescribed exercises/training | Comparator; adherence to prescribed exercises/training | Knee-specific proprioception test; outcome | Between-group (experimental vs control) comparisons of ACL-injured (reconstructed) limb – mean difference (95% CI)† |
|---|---|---|---|---|---|
| Zult et al (2018)[33] | Int: n=29 (22), 28±9 years; Com: n=26 (21), 28±10 years n=24 males n=20 females ACLR (patellar tendon graft/ hamstring tendon graft (SSG)/ Artificial) | Standard rehabilitation +Strength training of the quadriceps of the non-injured leg; two quadriceps exercises, 8–12 reps. maximum, 3 sets; two times/ week from week 1–12 after ACLR; **Adherence:** NR explicitly; however, one participant who performed <26 sessions was excluded from analysis after week 26 | Standard rehabilitation: 2 times/week from week 1–12 after ACLR; **Adherence:** NR explicitly; however, two participants who performed <26 sessions was excluded from analysis after week 26 | **Proprioception test 1: JPS¶** **Body position:** seated (?); **Instrument:** Biodex dynamometer (?); **Procedure:** passive-active, eyes blindfolded (?); **SA:** 90° (?); **TA:** 15°, 30°, 45°, and 60°; **Outcome measure:** AAE (one trial at each angle). **Proprioception test 2:** Quadriceps force control (QFC); **Body position:** seated (?); **Instrument:** Biodex dynamometer (?); **Procedure:** A target force matching task with the target set at 20% MVC for three isometric trials (at 65° of knee flexion (5 s duration)) and 40 Nm for dynamic trials (four concentric and eccentric trials at 20°/s from 10°–90° knee flexion) (20°/s between 10° and 90° of knee flexion); **Outcome measure:** force accuracy (absolute error) determined over the terminal 3 s data for isometric trials (at 65° knee flexion) and over the middle 2 s data for concentric and eccentric trials. | **JPS (°)** ** at 26 weeks post-ACLR:** **TA 15°:** 1.00(−1.12 to 3.12); **TA 30°:** 2.00(−0.12 to 4.12); **TA 45°:** −1.00(−3.39 to 1.39); **TA 60°:** −1.00(−2.79 to 0.79); **QFC (Nm)**†† at 6 months (26 weeks) post-ACLR:** **Concentric 60°/s:** 6.00(0.67 to 11.33); **Eccentric 60°/s:** −1.00(−3.99 to 1.99); **Isometric:** 1.00(−0.76 to 2.76) |

*Included in analysis.
†Calculated with Review Manager (RevMan) V.5.3 (The Cochrane Collaboration 2014, Nordic Cochrane Centre Copenhagen, Denmark).
‡Mean difference between groups were calculated based on postintervention/final follow-ups scores reported by the authors.
§Difference between four intervention groups and the comparator group were same and so only one comparison is presented.
¶JPS method has been presumed based on authors' reference to the method employed by Hortobágyi et al.[50]
**Mean difference between groups were calculated based on change scores from baseline (preintervention vs postintervention) reported by the authors.
††Quadriceps force accuracy; both legs (within each group) showed improved force control (22%–34%) at 26 weeks postsurgery (p<0.050) according to the authors.
ACLR, anterior cruciate ligament reconstruction; com, comparator group; Int, intervention group; JPS, joint position sense; NR, not reported; TTDPM, threshold to detection of passive motion.

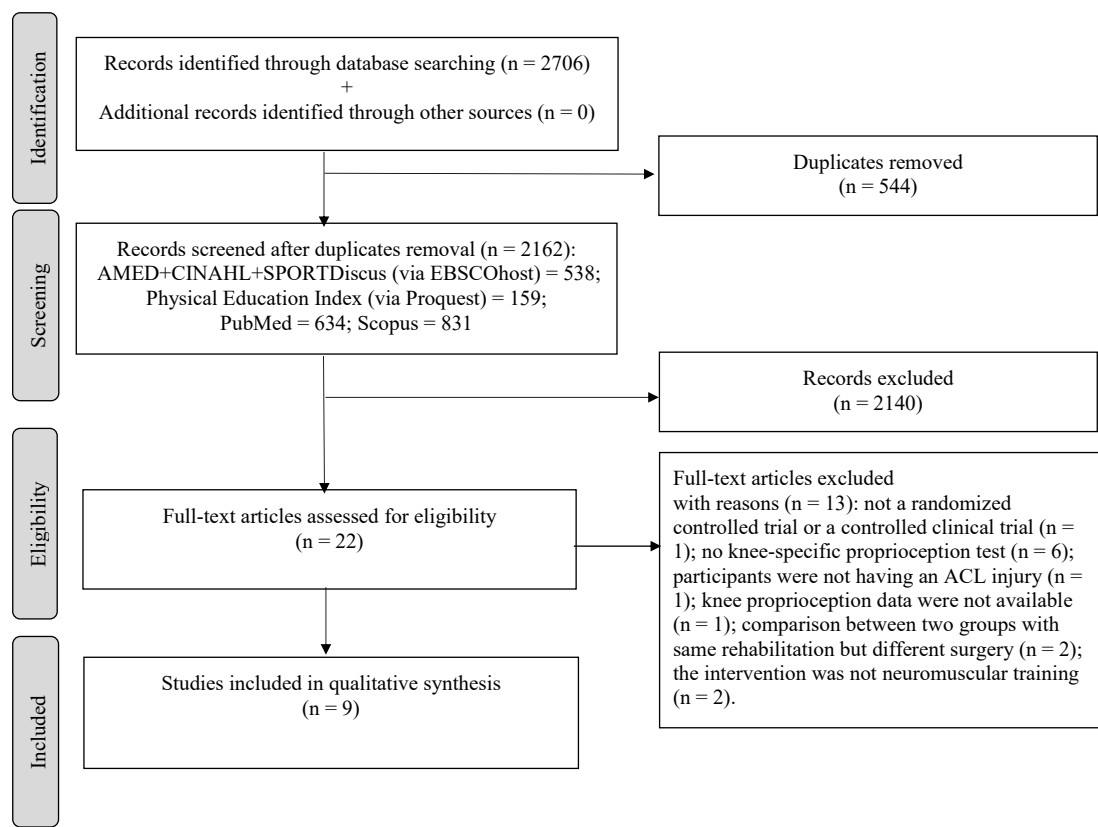

**Figure 1** Flow diagram depicting the steps involved in screening and selection of eligible articles. ACL, anterior cruciate ligament.

for four groups (at four inclination angles 0°, 5°, 10° and 15°, respectively) compared with standard rehabilitation in a comparator group.[31] Nevertheless, Zult et al examined the effects of cross-education of strength training of the non-injured leg along with standard rehabilitation compared with standard rehabilitation alone.[33]

### Knee-specific proprioceptive measures

Seven studies used active or passive JPS and all but one used (absolute) angular error (AAE) as a variable to evaluate the outcome.[28–34] Conversely, one study used a computer programme (monitored-rehab-system-software) to define a virtual line/route to allow joint repositioning within 30%–70% knee range of motion with and without visual feedback.[28] The differences between visual and blinded trials (two each) based on the deviations from the computer-generated line (in mm) were used to give information about the sense of proprioception.[28] All these studies used sitting or supine test position for assessing JPS. There were two to four predetermined target knee flexion angles across studies ranging from 15° to 80°.[29–34] Moreover, two studies[28 29] used active knee motion and four used passive knee motion[30–33] to set the target angle. Whether Cho et al used active or passive knee motion to set/reproduce the target angle seems ambiguous.[34] Four studies[28 29 32–34] used active knee motion and two[30 31] used passive knee motion to reproduce the target angle. The JPS method

used by Zult et al[33] was presumed based on their reference to Hortobágyi et al.[50]

The angular error was measured with 1–6 trials per each angle and one study[33] randomised the order of the joint angles used. Eyes were blinded during the test in six studies[29–34] while one study used visual feedback when the individual was placing the knee joint in the target angle but no such feedback was given during reproduction of the target angle.[28] The difference between visual and non-visual trials was calculated in mm by the device as a measure of JPS.[28] A Biodex dynamometer (Biodex Medical Systems, Shirly, New York, USA) was used in five studies[29 30 32–34] to test JPS. Even so, one study used a continuous passive motion equipment[31] while another[28] employed a functional squat system (Monitored Rehab System, Haarlem and the Netherlands) with a leg press machine and an associated computer programme for assessing JPS.

Three studies[31 48 49] evaluated knee kinesthesia with the TTDPM using a bespoke device,[48 49] or a continuous passive motion equipment.[31] The knee joint was moved in flexion or extension at a constant angular velocity of 0.5°/s[48] or 0.1°/s.[31 49] While the participants were blindfolded in two studies,[31 49] the other study did not mention about visual feedback.[48] In all three studies, the tests were performed three times in each direction (flexion and/or extension) for both legs but whether the order of direction or leg was randomised is not reported. In the study

**Table 3** Risk of bias assessment of included studies according to the revised Cochrane risk-of-bias tool for randomised trials (RoB 2)—judgements in five domains and an overall judgement using the descriptors of low risk of bias (low), some concerns and high risk of bias (High)

| Included studies | Outcome variable | 1. Bias from the randomisation process | 2. Bias due to deviations from intended interventions | 3. Bias due to missing outcome data | 4. Bias in measurement of the outcome | 5. Bias in selection of the reported result | Overall judgement |
|---|---|---|---|---|---|---|---|
| Baltaci *et al* 2013[28] | JPS | High | Some concerns | Low | High | High | High |
| Beynnon *et al* 2011[49] | TTDPM | Low | Low | Low | Low | Some concerns | Some concerns |
| Cho *et al* 2013[34] | JPS | Some concerns | Some concerns | Low | High | Some concerns | High |
| Fu *et al* 2013[30] | JPS | Low | Low | Low | Low | Some concerns | Some concerns |
| Kaya *et al* 2019[32] | JPS | Some concerns | High | High | Low | Some concerns | High |
| Moezy *et al* 2008[29] | JPS | Some concerns | Low | Low | Some concerns | High | High |
| Risberg *et al* 2007[48] | TTDPM | Low | Low | Low | Low | Some concerns | Some concerns |
| Shen *et al* 2019[31] | JPS | Some concerns | Low | Low | Low | Some concerns | Some concerns |
| | TTDPM | Some concerns | Low | Low | Low | Some concerns | Some concerns |
| Zult *et al* 2018[33] | JPS | Low | Some concerns | Low | High | Some concerns | High |
| | QFC | Low | Some concerns | Low | Low | Some concerns | Some concerns |

JPS, joint position sense; QFC, quadriceps force control; TTDPM, threshold to detect passive motion.

by Risberg *et al*.[48] TTDPM data were missing for 27 out of 74 participants because of device failure, which might lower the power of the study.

### Effects of NT on knee proprioception in individuals with ACLR

There were conflicting findings among the included studies for the effects of NT on improving JPS, TTDPM and QFC. Overall, mean differences between groups indicated inconsistent findings with an increase or decrease of JPS angular errors (one or more target angles) by ≤2°, TTDPM by ≤1.5°, and QFC (concentric/eccentric/isometric contractions) by ≤6 Nm following NT.

Of the nine included articles, four reported reduction in JPS angular errors of ACLR knee at one or more target angles (JPS at 45° but not 15°[34]; JPS at 60° but not 30°[29]; JPS at 15°, 45°, 75°[32]; JPS 20°, 50°, 80°[31] and/or contralateral non-injured knee (JPS at 30° and 60°)[29] favouring the NT group (exercises on a balance pad,[34] WBVT,[29] neuromotor control exercises[32] or backward treadmill walking.[31] Shen *et al* also reported improved TTDPM following backward treadmill walking.[31] When we calculated mean differences for author-reported postoperative[32] or change (preintervention vs postintervention) scores[29 34] between groups for the ACLR leg with the Review Manager V.5.3 software (the Cochrane Collaboration), their 95% CIs revealed no

effects (see table 2). Moreover, the remaining five studies did not report significant differences in proprioception between groups.[28 30 33 48 49]

### Assessing certainty in evidence

There were serious concerns with four GRADE domains (ROB, inconsistency, indirectness and imprecision associated with the findings) across the seven studies that measured JPS (tables 4 and 5). The certainty of evidence found was very low for the effects of NT on improving JPS following ACLR.

There were further serious concerns with four GRADE domains (ROB, inconsistency, indirectness and imprecision associated with the findings) across the three studies measuring TTDPM (tables 4 and 5). Therefore, the certainty of evidence found was very low for improving TTDPM in individuals with ACLR following NT (table 4).

An overall judgement of some concerns based on the Cochrane ROB 2 tool (table 3) was found for the study reporting changes in QFC following NT.[33] Available population, the magnitude and direction of effect, and effect estimates of QFC (tables 2 and 4) are derived from only one study which reflect serious concerns. However, the participants with ACLR, intervention (cross-education of the quadriceps with standard rehabilitation), and

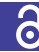

**Table 4** Applying the GRADE approach to rate the certainty in evidence found in the review

| No of studies | Study design | Certainty assessment | | | | | No of patients | | Certainty |
|---|---|---|---|---|---|---|---|---|---|
| | | Risk of bias | Inconsistency | Indirectness | Imprecision | Other considerations | Neuromuscular training | Comparator intervention | |
| **Knee joint position sense** | | | | | | | | | |
| 7 | Randomised trials | very serious* | serious† | serious‡ | serious§ | none | 139 | 105 | ⊕○○○ Very low |
| **Knee joint threshold to detect passive motion** | | | | | | | | | |
| 3 | Randomised trials | serious* | serious† | serious‡ | serious§ | none | 84 | 51 | ⊕○○○ Very low |
| **Quadriceps force control** | | | | | | | | | |
| 1 | Randomised trial | serious* | serious¶ | not serious | serious¶ | none | 22 | 21 | ⊕○○○ Very low |

GRADE domains are explained further in table 5.
*Included studies had a high RoB or some concerns based on the Cochrane ROB2 tool.
†The direction and/or magnitude of effect was inconsistent across trials.
‡Clinical heterogeneity (of participants, interventions, and method of assessing outcome measures).
§Number of participant <400 and/or wide 95% CIs of effect size estimates.
¶Available population, the magnitude and direction of effect, and effect estimates come from only one study.
GRADE, Grading of Recommendations, Assessment, Development and Evaluation; RoB, risk of bias.

QFC[33] are directly related to our research question. The certainty of evidence found was very low for improving QFC in individuals with ACLR following NT because only one relevant study was found.

## DISCUSSION

This review is the first, as far as we are aware, to address the effects of neuromuscular rehabilitation training on knee proprioception in individuals with ACL injury. A previous review, however, summarised the effects of proprioceptive and balance exercises following ACL injury/reconstruction on certain outcome measures (muscle strength, hop test, etc) but other than knee-specific proprioception tests.[51] Another similar review did not find any RCTs in this area.[52] We identified nine studies employing a range of NT methods, of which all but one[48] were published within the past decade. Nevertheless, there were serious concerns with two or more GRADE domains (ROB, inconsistency, indirectness or imprecision associated with the findings) across studies implying a very low certainty of evidence for improving JPS, TTDPM, and QFC of ACLR knee following NT.

### Effects of NT on knee proprioception in individuals with ACLR

Most of the employed NT programmes did not influence proprioception compared with comparator interventions. Potential reasons for insignificant between-group differences include: (1) experimental and comparator programmes (with exercises that are wholly or partly similar) which potentially might stimulate similar effects on proprioception in both programmes[28 30 32 34 48 49]; (2) the exercises did not adequately stimulate proprioception sense[33]; (3) a lack of proprioception deficit following ACL injury (TTDPM similar between ACL-injured and contralateral uninjured knee)[49]; (4) a lack of valid, sensitive and responsive knee-specific proprioception test methods; (5) a short follow-up period (a follow-up at least 18 months post-ACLR might be needed to regain proprioceptive function[53] in most studies except two studies[32 49]; (6) type II errors arising from low sample sizes in most studies (with missing power or sample size calculations); and (7) adherence rates of participants to the prescribed programme (only three studies have explicitly reported adherence rates to training sessions/exercises (table 2)).[30 48 49] The heterogeneity of interventions, methodological limitations, inconsistency in the magnitude and direction of effects, and imprecision of effect estimates, found in this review, preclude recommendation of one optimal NT intervention for improving proprioception following ACL injury in clinical practice.

### ROB in the included studies

Bias in selection of the reported variables/results due to absence of a prespecified plan of analyses applied to all but one study,[33] and none had published a trial protocol in a scientific journal although two studies were registered in a trial registry.[31 33] A possible reason for the absence of

**Table 5** GRADE evaluation of the certainty in evidence for knee joint position sense (JPS)

| GRADE domain | Reviewer judgement | Concerns about GRADE domains |
|---|---|---|
| **Knee JPS** | | |
| Risk of bias (methodological limitations) | Among seven RCTs[28–34] reporting changes in JPS following neuromuscular training, five RCTs were found to have a high risk of bias while the remaining two studies have some concerns based on the Cochrane ROB 2 tool (see table 3). Indeed, we judged that the included RCTs have very serious methodological limitations. | Very serious |
| Inconsistency | The direction and/or magnitude of effect on JPS was inconsistent across most of the included RCTs. In summary, the between-group comparisons of five RCTs showed borderline or no change in JPS angular errors of the ACLR knee for one or more target angles following interventions. We noted significant differences in reduction of JPS angular errors for all target angles favouring the intervention groups (backward treadmill walking or motor control exercises) in only two RCTs as reported by the authors.[31 32] In fact, Kaya et al had reported only postintervention scores but they neither reported nor compared the baseline scores (postoperative scores).[32] Two other studies[29 34] presented with insignificant effects at a low target angle (15° or 30°) and significant effects at a high target angle (45° or 60°) of JPS favouring the intervention group (whole-body vibration therapy[29] or exercises on a balance pad.[34] When we calculated mean differences for author-reported postoperative[32] or change (preintervention vs postintervention) scores,[29 34] between groups for the ACLR leg with the Review Manager V.5.3 software (the Cochrane Collaboration), their 95% CIs revealed no effects. Overall, we judged the evidence to have serious inconsistency in the direction and/or magnitude of effects. | Serious |
| Indirectness | The participants (with ACLR (different grafts)), different neuromuscular training and comparator interventions, and knee specific JPS measures in the included studies provide evidence to the research question. However, the heterogeneity of interventions precludes recommendation of one optimal neuromuscular training intervention for clinical practice. In addition, variations in the methods of JPS measurements (active vs passive angle reproduction, low vs high target angles, etc) precluded a meta-analysis. We judged the evidence to have serious indirectness especially owing to variations in the interventions and outcome measures. | Serious |
| Imprecision | A total of 244 patients was included from seven RCTs reporting changes in JPS following neuromuscular training (n=139) or comparator interventions (n=105). Most of the included trials reported non-significant results with wider 95% confidence intervals for one or more JPS (target) angles (see table 2). Therefore, we judged the evidence to have serious imprecision. | Serious |
| Other considerations | Since negative and positive findings have been published, and a comprehensive search for RCTs has been done, we did not suspect a publication bias. | None |
| **Knee joint TTDPM** | | |
| Risk of bias (methodological limitations) | Three RCTs[31 48 49] reporting changes in TTDPM following neuromuscular training were found to show some concerns in risk of bias based on the Cochrane ROB 2 tool (see table 3). We judged the included RCTs to be of serious methodological limitations. | Serious |
| Inconsistency | The direction and/or magnitude of effect was conflicting between the three RCTs. As two trials reported insignificant effects and one[41] reported significant effects (see table 2), we judged the evidence to have serious inconsistency in the direction and/or magnitude of effects. | Serious |
| Indirectness | The participants (with ACLR (different grafts)), different neuromuscular training and comparator interventions, and knee specific TTDPM measures in the included studies provide some evidence to the research question in hand. However, the heterogeneity of interventions and TTDPM measurements (starting angles, angular velocity, etc) precluded a meta-analysis. We judged the evidence to have serious indirectness especially owing to variations in the interventions and TTDPM methods. | Serious |

Continued



**Table 5** Continued

| GRADE domain | Reviewer judgement | Concerns about GRADE domains |
|---|---|---|
| Imprecision | A total of 135 patients was included in three RCTs reporting the effects of neuromuscular training (n=84) or comparator interventions (n=51) on TTDPM. Two trials[48 49] reported non-significant results while another one[31] reported significant effects which is evident with their confidence intervals (see table 2). However, Shen *et al* reporting significant effects on TTDPM included only 10 to 11 participants in each group while the other two studies with a relatively larger sample size declared no significant effects on TTDPM. Therefore, we judged the evidence to have serious imprecision. | Serious |
| Other considerations | As both negative and positive findings have been published, and a comprehensive search for RCTs has been done, we did not suspect a publication bias. | None |

ACLR, anterior cruciate ligament reconstruction; RCTs, randomised controlled trials; ROB, risk of bias; TTDPM, thresholds to detect passive motion.

registration for most studies in this review may be that all but three studies were older than 5 years. Yet, one latest published study did not report trial registration.[32]

Another concern was the method used to measure JPS. For instance, estimates of JPS based on 3–5 repetitions, in clinical trials, may be insufficient.[54] According to Selfe *et al* five repetitions in active knee JPS test, and six when performed passively, are necessary to ensure a consistent proprioception score.[55] However, this was only met in two included studies.[29 32]

All studies used AAE for measuring JPS acuity which represents a task-oriented approach to studying performance skill, in contrast to a process-orientation in which underlying processes are in focus. The inconsistency in performance, that is, response variability (variable error), may reflect noise in sensory signal and its processing[56] and thus be a more process-oriented outcome than AAE. To understand possible underlying mechanisms, it would be advantageous to combine task-oriented and process-oriented measures.

In general, method descriptions of proprioception tests were short and, in some studies, deficient, lacking information about factors that could influence the results. One such factor was randomisation of the order of target positions (cf. Zult *et al*),[33] which is required to minimise the effect of memory and reduce motor elements of the test. This is particularly applicable in tests with active positioning, which was the case for most studies, enabling central motor programmes.[57] Inadequate reporting of the proprioception tests would hinder their replication and raise ROB rating. Moreover, Kaya *et al* reported only post-intervention JPS scores, precluding baseline scores, despite claiming their study to be an RCT.[32]

Among seven RCTs[28–34] investigating changes in JPS following NT, five RCTs were found to have a high ROB while the remaining two studies have some concerns based on the Cochrane ROB 2 tool (table 3). Therefore, included RCTs have been judged to have very serious methodological limitations in the GRADE evidence synthesis.

## Mechanisms underpinning NT following ACLR

Two of the included studies evaluated the effects of WBVT[29 30]; however, only one found a favourable effect on proprioception (JPS—target angle 60°).[29] Two factors may contribute to the different findings between these studies. First, time point at which WBVT was given: Fu *et al* employed WBVT at 1 month post-ACLR for 2 months and evaluated JPS at 3 and 6 months after the surgery (table 3).[30] On the other hand, Moezy *et al* gave WBVT at 3 months post-ACLR for 1 month and assessed JPS at 4 months after the surgery.[29] It seems starting WBVT at 3 months, rather than at 1 month, post-ACLR might have better on improving knee JPS. Second, the use of active[29] or passive[30] knee movement when testing JPS. Active tests stimulate both joint and muscle-tendon mechanoreceptors and induce alpha-gamma coactivation while passive tests assess joint receptors to a higher degree[10 58] which potentially could mean a higher sensitivity of the active test.

WBVT has shown effects on body posture, flexibility, proprioception (TTDPM in patients with osteoarthritis), coordination and muscle power.[59–61] It has been promoted as an effective method to induce a reflex muscle contraction in subjects with difficulties to evoke voluntary contractions.[62] The mechanism behind the improvements may be that the vibration stimuli excite muscle spindles, and activate the tonic vibration reflex, which acts on alpha-motor neurons. This could potentially engage central motor command, which facilitates increased muscle activation and voluntary movements.[59]

Cho *et al* showed a significant effect on knee proprioception (JPS and TTDPM) with closed kinetic chain exercises on a balance pad/board.[34] Exercises on a balance board are widely used to improve proprioception.[38 51] In this review, a few NT programmes included, among other exercises, balance training with or without a balance pad/board.[28 32 34 48 49] Additionally, one study claimed backward walking, a closed kinematic chain exercise, to stimulate joint/muscle receptors and sensory afferents to the CNS and augment proprioceptive and balance training.[31]

Among these studies, all but one,[31] did not show significant mean differences between groups in proprioception calculated using the Review Manager V.5.3 software (the Cochrane Collaboration) (see table 2 and online supplemental file). Different designs and levels of difficulty of the execution were found (eg, a simple static balance task (with and without visual input), dynamic exercises performed on the balance board, backward walking on a treadmill, etc).

There is a challenge to transfer the rehabilitation in the clinic to automatic movements required for athletic activities.[18 63] Wii Fit or similar games have the potential to combine feedback with an external focus in a sport-specific environment,[28] supporting the use of such training tools. However, a study on Nintendo Wii Fit training did not support its use for improving knee proprioception following ACLR.[28] Newer technology with stroboscopic eyewear might have the potential to decrease visual input without fully occluding it, making it possible to use them in sport specific rehabilitation. To prepare the individual for complex athletic environments and reduce reinjury risk, rehabilitation might focus on NT with reduced demands on visual inputs and enhance automatic movement control with cognitive demands included.[18] Whether such NT training improves knee proprioception and, how this should be assessed in the best way,[13] are yet to be determined.

### The ability of tests to discern changes in proprioception following NT

There is neither a gold-standard proprioception test (targeting JPS, kinesthesia, force sense) nor a standard procedure with established psychometric properties to test each proprioception sense following ACL injury. In this review, JPS and TTDPM were commonly reported. The Ruffini and Golgi receptors are slow-adapting receptors, responding to a change in joint position. Nevertheless, the Pacinian receptors that respond to low degrees of joint stress are more sensitive to rapid changes in accelerations and contribute to a low TTDPM.[2 64] JPS has been reported to detect a greater difference in knee proprioception than TTDPM following an ACL injury.[2] However, our findings remain equivocal regarding the outcomes of JPS or TTDPM following NT.

Knee-specific proprioception tests provide an indirect measure of proprioception involving the process of the CNS.[10] Psychosocial factors,[65] pain and preinjury motor skills may influence the central mechanisms and the outcome of such tests following NT. Knee-specific proprioception tests are designed to exclude motor skills, but how successful that exclusion works, remains unclear.

### Limitations and future recommendations

The nine included studies looked at only individuals with ACLR but not those managed conservatively following ACL injury. Owing to clinical heterogeneity of interventions and outcome measurements, meta-analyses were precluded from the GRADE evidence synthesis. The included studies had methodological limitations (high ROB or some concerns) and all, but two studies,[31 33] had not preregistered/published their protocol. There is a need for high-quality RCTs with low ROB in this area.

Grey literature was not included in the current review which could be seen as a limitation. The most common reason for exclusion of clinical trials in this review was that they did not evaluate the effects of NT following ACLR with a knee-specific proprioception test. Perhaps, the lack of consensus regarding the most appropriate, valid, reliable and responsive proprioception tests, number of target angles or most responsive target angles (low vs high) might have precluded such outcomes in these studies. Therefore, psychometric properties of such tests must be established.[13]

When designing rehabilitation programmes with long-term follow-up, aberrations in neuromotor control as well as neuroplastic changes should preferably be addressed. To reflect a wide spectrum of individual impairments, further research should investigate differences in individuals with ACL injuries managed with surgical (graft types) or conservative treatment, both sexes, athletes and non-athletes of different ages. Future studies might assess neuromotor control in functional tasks rather than relying on knee-specific proprioception tests, given the challenges of isolating the proprioceptive ability.

### CONCLUSION

The existing nine studies on individuals with an ACLR using heterogeneous interventions and knee-specific proprioception measures revealed a very low certainty in current evidence for employing NT programmes to improve knee proprioception. The GRADE evidence synthesis revealed a high ROB or some concerns, indirect evidence, conflicting findings and imprecision of effect estimates in the included studies. Well-designed RCTs with homogeneous populations (having ACL injury managed with or without reconstruction), novel/well-designed NT interventions and valid proprioception measures are warranted to substantiate conclusive evidence in this area.

**Contributors** AA and CKH conceived the idea of the project. AA, MB, SM and CKH were responsible for designing the review and conceptualising the initial review protocol. AA led the writing of the manuscript. MB, SM and CKH contributed to writing the manuscript. AA, MB and CKH have reviewed and revised the manuscript for intellectual content. All authors approved the final version of the manuscript. AA is the guarantor of this work.

**Funding** The work was supported by the Swedish Research Council (2017–00892); Region Västerbotten (ALF 7003575; Strategic funding VLL-358901; Project. No. 7002795); the Swedish Research Council for Sports Science (CIF P2019 0068) and King Gustaf V and Queen Victoria's Foundation of Freemasons 2019 (Häger).

**Disclaimer** The funders were not involved in the conception, design, execution and writing of the review.

**Competing interests** None declared.

**Patient consent for publication** Not required.

**Provenance and peer review** Not commissioned; externally peer reviewed.

**Data availability statement** All data relevant to the study are included in the article or uploaded as online supplemental information.

**ORCID iD**
Ashokan Arumugam http://orcid.org/0000-0001-5795-3812

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
