## [Reviewer comments · BMJ Open]

ARTICLE DETAILS

TITLE (PROVISIONAL)	Effects of neuromuscular training on knee proprioception in individuals with anterior cruciate ligament injury - A systematic review and GRADE evidence synthesis
AUTHORS	Arumugam, Ashokan; Björklund, Martin; Mikko, Sanna; Häger, Charlotte

VERSION 1 – REVIEW

REVIEWER	Seixas, Adérito Fundação Ensino e Cultura Fernando Pessoa
REVIEW RETURNED	10-Feb-2021

GENERAL COMMENTS	Review Report The authors have submitted a manuscript aiming to systematically review and summarize the evidence analysing the effects of neuromuscular training (NT) on knee proprioception in individuals following ACL injury. I congratulate the authors for their interesting and relevant work, which has the potential to be an interesting contribution to BMJ Open. There are, however, some issues that must be addressed before publication, which will be covered in this review report. Abstract: No information regarding the meta-analysis is provided. Introduction The introduction is well structured, the research problem is well framed and a clear objective is provided in the section. Methods In general, the methodology is scientifically sound. However, I have some concerns regarding the following specific aspects. The eligibility criteria subsection must be improved regarding the type of studies included in the review. Regarding the data sources, a systematic review aims to identify, evaluate and summarise the evidence of all relevant individual studies but the authors do not provide information regarding their efforts to include grey (unpublished) literature, which is a well-known issue in systematic reviews. Grey literature can be an important source of studies for inclusion in reviews (Higgins et al., 2019). Please clarify on this point as the non-inclusion, or lack of efforts to find relevant grey literature is a limitation of the systematic review and must be stated as such in the limitations section. The authors state that they did not suspect a publication bias but if no efforts to locate relevant
---

	grey literature were not conducted, this may not be the case. In the evidence synthesis subsection please provide a more detailed description regarding the meta-analytic process, e.g. the chosen model/method, effect measures and calculation (difference in scores/final scores). Why haven't the authors included a publication bias assessment in the meta-analysis? Although not consensual, please consider reporting the results for publication bias (hypothesis testing and visual inspection of the funnel plot) or provide the rationale not to. Why haven't the authors explored the sources of heterogeneity, at least in the JPS studies? It would be interesting to conduct a subgroup analysis of passive versus active repositioning and levels of risk of bias (some concerns versus high risk). Maybe it would be adequate to explore univariate meta-regression with joint repositioning angle and follow-up duration as univariate moderators? Results The PRISMA Flow Diagram may need some adjustments. The authors could provide the results for each included database and then the combined results (2706), only then the number of duplicated results should appear. As it is it seems that 544 results were removed from the 2162, and then the same number appears in the following box. This should be clarified. Moreover, in the reasons for exclusion of full-texts, the reason "not an RCT" should in fact be "not an RCT or CT". Please include a subsection for the meta-analysis reporting heterogeneity when adequate and that, considering moderate/high heterogeneity (reporting the value), the meta-analytic measures must be interpreted carefully, or not reported at all, but please describe the rationale for your options. Again, congratulations for your work. References: Higgins, J. P., Thomas, J., Chandler, J., Cumpston, M., Li, T., Page, M. J., & Welch, V. A. (Eds.). (2019). Cochrane handbook for systematic reviews of interventions. John Wiley & Sons.
--	---

REVIEWER	Moore, Joe University of Portsmouth
REVIEW RETURNED	10-Feb-2021

GENERAL COMMENTS	Overall Comments Firstly, congratulations must go to the authors on conducting such a review where the time invested must have been high. I think the overall topic of the review is excellent and the discussion does a nice job of highlighting some of the possible mechanisms for the identified conclusions. My concerns are around the data analysis section where a lack of detail and clarity means as a reader I found it difficult to determine how you came to a conclusion around whether the NT was favourable or not. The article is well written although seems to mix
---

	perspectives (1st and 3rd person), and there are a few sections that I think could be tightened up a little to avoid repetition. Specific Comments Line 30-31 – “The low evidence thus questions common clinical neuromuscular training programs in practice”. I am not convinced your evidence supports this conclusion. Yes the evidence questions the suitability of NT for improving proprioception, but as you do not evaluate the other benefits of NT (movement patterns, dynamic stability) I think this sentence needs rewording. Line 57 – young seems a very broad term, perhaps an age range in brackets for clarity Line 60 – include reference to reconstruction as the predominant treatment Line 167 – change to third person to be consistent with the rest of the text Line 166-172 – I find this section lacking information and a little confusing. For example throughout the results and discussion the authors discuss that NT has (or does not have) a favourable effect on proprioception but it is not clear how this result was determined. This may be as I am not familiar with the GRADE method, but as this is the main analysis conducted a more thorough explanation is required. Line 171-172 state no meta-analyses were conducted and I agree with the justification, but then meta-analyses are provided in the supplementary material. I would advise these to be included or removed completely. It may be possible to report the mean differences to provide details of the effects but not combine these to create a meta-analysis due to the issues you have raised.. Lines 210 – 230 – I appreciate the difficulty reporting the varying ranges of rehabilitation plans conducted within the identified articles, however I am unsure what this paragraph adds beyond what is reported in Table 1. Perhaps consider condensing to focus on presenting the different “types” of NT (e.g. unstable surface, balance tasks, plyometrics) in the literature rather than simply listing each article separately. Line 315-338: The risk of bias is clearly moderate to high in all studies. Whilst the description is detailed, I think some insight into how this bias impacts the conclusions that can be drawn from the evidence. Lines 342-343: Perhaps include insight into which timing demonstrated a favourable effect
--	--

VERSION 1 – AUTHOR RESPONSE

Reviewer: 1

Dr. Adérito Seixas, Fundação Ensino e Cultura Fernando Pessoa

Comments to the Author:

Review Report

The authors have submitted a manuscript aiming to systematically review and

summarize the evidence analysing the effects of neuromuscular training (NT) on knee proprioception in individuals following ACL injury. I congratulate the authors for their interesting and relevant work, which has the potential to be an interesting contribution to BMJ Open.

There are, however, some issues that must be addressed before publication, which will be covered in this review report.

Abstract:

No information regarding the meta-analysis is provided.

Reply: Thank your for your constructive feedback. Regarding the meta-analysis, please see our response above to the editor. We have decided to withdraw the meta-analysis based on too large clinical heterogeneity of interventions and outcome measurements in addition to substantial statistical heterogeneity ($I^2 > 60\%$).

Introduction

The introduction is well structured, the research problem is well framed and a clear objective is provided in the section.

Methods

In general, the methodology is scientifically sound. However, I have some concerns regarding the following specific aspects. The eligibility criteria subsection must be improved regarding the type of studies included in the review.

This has now been clarified (line 143). Though we were interested in both randomized controlled trials and controlled trials, we found that only nine randomized controlled trials were eligible for inclusion in the review.

Revision: Lines 143

5. Study design: randomized controlled trials (RCTs) or controlled clinical trials. Regarding the data sources, a systematic review aims to identify, evaluate and summarise the evidence of all relevant individual studies but the authors do not provide information regarding their efforts to include grey (unpublished) literature, which is a well-known issue in systematic reviews. Grey literature can be an important source of studies for inclusion in reviews (Higgins et al., 2019). Please clarify on this point as the non-inclusion, or lack of efforts to find relevant grey literature is a limitation of the systematic review and must be stated as such in the limitations section. The authors state that they did not suspect a publication bias but if no efforts to locate relevant grey literature were not conducted, this may not be the case.

Reply: We understand the concern of the reviewer and certainly agree that grey literature has been excluded in our databases search. Now we have added this as a limitation of our systematic review (line 454).

While employing a pre-designed comprehensive search strategy in electronic databases, in accordance with our PROSPERO protocol (CRD42018107349) based on well-defined eligibility criteria for studies, in six renowned databases and screening 2700 retrieved citations, only nine RCTs met the eligibility criteria to address the research question. We did not find any controlled clinical trial matching our inclusion criteria. Since both positive and negative findings have been published and we did a comprehensive search of the literature, we felt that a publication bias is less likely to influence overall certainty of evidence reported in this review. Nevertheless, the overall certainty of evidence for all

the outcomes of interest of knee proprioception following neuromuscular training was found to be very low and even if we include publication bias under “other considerations” in the GRADE synthesis (Tables 3 and 4), the certainty of evidence will still remain very low. We expect that adding a few more unpublished studies (if there are any) is less likely to change the certainty of evidence found in the review.

Based on the number of RCTs found eligible (n=9) and the risk of bias associated with them, it clearly transpires that methodologically robust RCTs are further warranted to substantiate the effects of neuromuscular training on knee proprioception measures following ACL reconstruction.

We are unable to do funnel plots with trim and fill analyses¹⁰ to help identify and correct for the presence of publication bias because it was considered appropriate only when the following criteria described by Ioannidis and Trikalinos¹¹ were met: the inclusion of at least ten studies (with statistically significant results in at least one of the included studies), statistical heterogeneity < 50% and the ratio of the maximal to minimal variance across studies > 4.

Revision: Line 454

“Grey literature was not included in the current review, which could be seen as a limitation.”

In the evidence synthesis subsection please provide a more detailed description regarding the meta-analytic process, e.g. the chosen model/method, effect measures and calculation (difference in scores/final scores). Why haven't the authors included a publication bias assessment in the meta-analysis? Although not consensual, please consider reporting the results for publication bias (hypothesis testing and visual inspection of the funnel plot) or provide the rationale not to.

Why haven't the authors explored the sources of heterogeneity, at least in the JPS studies? It would be interesting to conduct a subgroup analysis of passive versus active repositioning and levels of risk of bias (some concerns versus high risk). Maybe it would be adequate to explore univariate meta-regression with joint repositioning angle and follow-up duration as univariate moderators?

Reply: We thank the reviewer for the suggestions. We have, however, decided to exclude subgroupings and meta-analysis due to the widely varying methodology between studies. The difference between JPS tests in the included studies is thoroughly summarized in lines 268-290. Regarding JPS studies, at most two studies used the same method (active-active,^{2,5} passive-passive^{4,7} or passive-active^{3,9}) but the starting and target angles used varied between these knee proprioception tests in the included studies, if they were reported at all. And above that, studies were also heterogeneous concerning the number of trials per each angle. Further, the neuromuscular training interventions, targeting JPS, widely varied between studies^{1-5,7,9} (please see lines 242-266 and Table 1): closed kinetic chain exercises on a balance pad,¹ whole-body vibration therapy (WBVT),^{2,7} neuromuscular (motor control) exercises for the lower limbs,³ backward walking on a treadmill,⁴ Nintendo Wii Fit training,⁵ and cross-education of strength training of the non-injured leg along with standard rehabilitation.⁹ Indeed, there was a substantial heterogeneity (JPS: I²=67%; TTDPM: I²=86%; QFC: I²=61%) when we performed a meta-analysis using a random-effects model for the outcomes of interest.

We interpreted I² statistic 50% to 90%: may represent substantial heterogeneity -

(<https://training.cochrane.org/handbook/current/chapter-10#section-10-10-2>)

To combine values of studies with large methodological differences and inconsistent findings (with conflicting direction of effects) may create spurious conclusions. Further, the Cochrane team recommends to exclude meta-analysis – “If there is considerable variation in results, and particularly if there is inconsistency in the direction of effect, it may be misleading to quote an average value for the intervention effect.”

(<https://training.cochrane.org/handbook/current/chapter-10#section-10-10-3>)

We have added the required explanation in the main text to make it explicit for the readers.

Revision: Lines 188-202

“A meta-analysis was precluded owing to clinical heterogeneity of interventions and outcome measurements (JPS, TTDPM and QFC). For instance, despite seven studies targeting JPS, a meta-analysis was not appropriate because at most two studies used the same method (active-active,^{28,29} passive-passive^{30,31} or passive-active^{32,33}) but the starting and target angles and the number of trials per each angle varied between these proprioception tests in the included studies. Further, the neuromuscular training interventions, targeting JPS, widely varied between studies²⁸⁻³⁴: closed kinetic chain exercises on a balance pad,³⁴ whole-body vibration therapy (WBVT),^{29,30} motor control exercises for the lower limbs,³² backward walking on a treadmill,³¹ Nintendo Wii Fit training,²⁸ and cross-education of strength training of the non-injured leg along with standard rehabilitation.³³ Further, in addition to inconsistent findings among the studies, a significant statistical heterogeneity ($I^2 > 60\%$) in a random-effects meta-analysis was evident. Although meta-analyses were excluded, the Review Manager 5.3 software (the Cochrane Collaboration) was used to calculate between-group mean differences (effect sizes) and their 95% confidence intervals for summarizing the findings for each outcome of interest in Table 1.”

Results

The PRISMA Flow Diagram may need some adjustments. The authors could provide the results for each included database and then the combined results (2706), only then the number of duplicated results should appear. As it is it seems that 544 results were removed from the 2162, and then the same number appears in the following box. This should be clarified. Moreover, in the reasons for exclusion of full-texts, the reason “not an RCT” should in fact be “not an RCT or CT”.

Reply: We have amended the PRISMA flowchart (Figure 1) as suggested.

Please include a subsection for the meta-analysis reporting heterogeneity when adequate and that, considering moderate/high heterogeneity (reporting the value), the meta-analytic measures must be interpreted carefully, or not reported at all, but please describe the rationale for your options.

Again, congratulations for your work.

References:

Higgins, J. P., Thomas, J., Chandler, J., Cumpston, M., Li, T., Page, M. J., & Welch, V. A. (Eds.). (2019). *Cochrane handbook for systematic reviews of interventions*. John Wiley &

Sons.

Reply: We thank the reviewer, again, for the congratulations. We have now elaborated the reasons for exclusion of the meta-analysis in lines 188-202 (please see our response above).

Reviewer: 2
Dr. Joe Moore, University of Portsmouth

Comments to the Author:
Overall Comments

Firstly, congratulations must go to the authors on conducting such a review where the time invested must have been high. I think the overall topic of the review is excellent and the discussion does a nice job of highlighting some of the possible mechanisms for the identified conclusions. My concerns are around the data analysis section where a lack of detail and clarity means as a reader I found it difficult to determine how you came to a conclusion around whether the NT was favourable or not. The article is well written although seems to mix perspectives (1st and 3rd person), and there are a few sections that I think could be tightened up a little to avoid repetition.

Reply: We thank the reviewer for their comments and the congratulations. Please see our comment-by-comment responses below.

Specific Comments

Line 30-31 – “The low evidence thus questions common clinical neuromuscular training programs in practice”. I am not convinced your evidence supports this conclusion. Yes the evidence questions the suitability of NT for improving proprioception, but as you do not evaluate the other benefits of NT (movement patterns, dynamic stability) I think this sentence needs rewording.

Thank you for pointing this out. We have now removed this sentence.

Line 57 – young seems a very broad term, perhaps an age range in brackets for clarity

Reply: We have now added the required information.

Revision: Line 58

“.....(14-18 years for females and 19-25 years for males).12”

Line 60 – include reference to reconstruction as the predominant treatment

We have included reference no. 3 to denote ACL reconstruction as the predominant treatment:

Sanders TL, Maradit Kremers H, Bryan AJ, et al. Incidence of anterior cruciate ligament tears and reconstruction: a 21-year population-based study. Am J Sports Med. 2016;44(6):1502-1507.

Revision: Line 62-63

“.....ACL injury is predominantly treated by surgical reconstruction,3.....”

Line 167 – change to third person to be consistent with the rest of the text

Reply: Changed accordingly as suggested (line 177).

Line 166-172 – I find this section lacking information and a little confusing. For example throughout the results and discussion the authors discuss that NT has (or does not have) a favourable effect on proprioception but it is not clear how this result was determined. This may be as I am not familiar with the GRADE method, but as this is the main analysis conducted a more thorough explanation is required.

Reply: We agree that it might be difficult for readers to understand this section if they are not familiar with the GRADE domains. We have included the required information as suggested:

Revision: Lines 176-187

“The overall evidence level in this review was determined using the Grading of Recommendations, Assessment, Development and Evaluation (GRADE) tool considering the following five domains: 1. risk of bias: high risk, some concerns, or low risk associated with knee proprioception measures based on the Cochrane ROB 2 tool; 2. Inconsistency of findings: similar or conflicting direction of effect, effect estimates and overlap of confidence intervals for knee proprioception measures from different studies; 3. indirectness of evidence: appropriateness of participants, interventions, and outcomes used to answer the review question; 4. imprecision of results: the length of 95% confidence intervals of effect estimates and overall sample (number of participants) from which effect estimates are derived; and other domains: e.g. publication bias if applicable.²⁷ The overall evidence was rated as very low, low, moderate or high.”
Line 171-172 state no meta-analyses were conducted and I agree with the justification, but then meta-analyses are provided in the supplementary material. I would advise these to be included or removed completely. It may be possible to report the mean differences to provide details of the effects but not combine these to create a meta-analysis due to the issues you have raised..

Reply: We certainly agree with the reviewer. We have now decided to exclude the meta-analysis from online supplementary material and have stated the reasons for excluding such analysis in the manuscript. We have retained the effect sizes for knee proprioception measures mentioned in Table 1 and added a brief section in the results too (as suggested by the editor).

Revision:

lines 188-202

“A meta-analysis was precluded owing to clinical heterogeneity of interventions and outcome measurements (JPS, TTDP and QFC). For instance, despite seven studies targeting JPS, a meta-analysis was not appropriate because at most two studies used the same method (active-active,^{28,29} passive-passive^{30,31} or passive-active^{32,33}) but the starting and target angles and the number of trials per each angle varied between these proprioception tests in the included studies. Further, the neuromuscular training interventions, targeting JPS, widely varied between studies²⁸⁻³⁴: closed kinetic chain

exercises on a balance pad,34 whole-body vibration therapy (WBVT),29,30 motor control exercises for the lower limbs,32 backward walking on a treadmill,31 Nintendo Wii Fit training,28 and cross-education of strength training of the non-injured leg along with standard rehabilitation.33 Further, in addition to inconsistent findings among the studies, a significant statistical heterogeneity ($I^2 > 60\%$) in a random-effects meta-analysis was evident. Although meta-analyses were excluded, the Review Manager 5.3 software (the Cochrane Collaboration) was used to calculate between-group mean differences (effect sizes) and their 95% confidence intervals for summarizing the findings for each outcome of interest in Table 1.”

Lines 300-316

“Effects of NT on Knee Proprioception in Individuals with ACLR

There were conflicting findings among the included studies for the effects of NT on improving JPS, TTDPM and QFC. Overall, mean differences between groups indicated inconsistent findings with an increase or decrease of JPS angular errors (one or more target angles) by $\leq 2^\circ$, TTDPM by $\leq 1.5^\circ$, and QFC (concentric/eccentric/isometric contractions) by ≤ 6 Nm following neuromuscular training.

Of the nine included articles, four reported reduction in JPS angular errors of ACLR knee at one or more target angles (JPS at 45° but not 15° 34; JPS at 60° but not 30° 29; JPS at 15° , 45° , 75° 32; JPS 20° , 50° , 80° 31) and/or contralateral non-injured knee (JPS at 30° and 60° 29) favoring the NT group (exercises on a balance pad 34, whole-body vibration therapy 29, neuromotor control exercises 32 or backward treadmill walking 31). Shen et al. also reported improved TTDPM following backward treadmill walking.31 When we calculated mean differences for author-reported post-operative 29,32 or change (pre- vs. post-intervention) scores 34 between groups for the ACLR leg with the Review Manager 5.3 software (the Cochrane Collaboration), their 95% confidence intervals revealed no effects (see Table 1). Moreover, the remaining five studies did not report significant differences in proprioception between groups.28,30,33,48,49”

Lines 210 – 230 – I appreciate the difficulty reporting the varying ranges of rehabilitation plans conducted within the identified articles, however I am unsure what this paragraph adds beyond what is reported in Table 1. Perhaps consider condensing to focus on presenting the different “types” of NT (e.g. unstable surface, balance tasks, plyometrics) in the literature rather than simply listing each article separately.

Reply: With all respect, we understand the concern of the reviewer. However, owing to a multitude of interventions, it was difficult for us to classify the interventions under different explicit categories. Moreover, most studies had multi-intervention training. We have tried to report studies together or one after the other if they had one primary intervention that was similar (e.g. whole body vibration therapy,2,7 Balance training using a balance pad/wobble board1,8).

There have been recent recommendations on using an external focus, rather an internal focus, for the acquisition or control of complex motor skills needed for sports. We have briefly highlighted this in the discussion (please see lines 421-430). Now we have highlighted that study by Baltaci et al. focused on Nintendo Wii Fit training to combine feedback with an external focus in a sport-specific environment while all others studies seem to have relied on internal focus for training under the results.

Revision: Lines 244-248

“Only one study by Baltaci et al. investigated the effects of using feedback with an external focus in a simulated sport-specific gaming environment with Nintendo Wii Fit compared to conventional rehabilitation.²⁸ On the contrary, the remaining eight studies focused on having an internal focus (mainly related to the position of specific body parts) for neuromuscular training.”

Line 315-338: The risk of bias is clearly moderate to high in all studies. Whilst the description is detailed, I think some insight into how this bias impacts the conclusions that can be drawn from the evidence.

GRADEpro software recommends that Cochrane risk of bias assessments may be used directly to inform the assessment of study limitations in the GRADE approach. In particular:

- low risk of bias would indicate “no serious limitations”
- unclear risk of bias (some concerns) would indicate either “no serious limitations” or “serious limitations”
- high risk of bias would indicate either “serious limitations” or “very serious limitations”

Revision: Lines 386-389

“Among seven RCTs²⁸⁻³⁴ investigating changes in JPS following NT, five RCTs were found to have a high risk of bias while the remaining two studies have some concerns based on the Cochrane ROB 2 tool (Table 2). Therefore, included RCTs have been judged to have very serious methodological limitations in the GRADE evidence synthesis.”

Lines 342-343: Perhaps include insight into which timing demonstrated a favourable effect

Reply: Amended as suggested.

Revision – Lines 393-398

“First, time point at which WBVT was given: Fu et al. employed WBVT at one-month post-ACLR for 2 months and evaluated JPS at 3 and 6 months after the surgery (Table 2).³⁰ On the other hand, Moezy et al. gave WBVT at 3 months post-ACLR for one-month and assessed JPS at 4 months after the surgery.²⁹ It seems starting WBVT at 3 months, rather than at one-month, post-ACLR might have better on improving knee JPS.”

We thank the editor and reviewers, again, for their constructive feedback. We hope that we have adequately addressed all of the comments and concerns raised and that the revised manuscript now meets your approval.

1. Cho SH, Bae CH, Gak HB. Effects of closed kinetic chain exercises on proprioception and functional scores of the knee after anterior cruciate ligament reconstruction. *Journal of Physical Therapy Science*. 2013;25(10):1239-1241.

2. Moezy A, Olyaei G, Hadian M, Razi M, Faghihzadeh S. A comparative study of whole body vibration training and conventional training on knee proprioception and postural stability after anterior cruciate ligament reconstruction. *British journal of sports Medicine*. 2008;42(5):373-385.
3. Kaya D, Guney-Deniz H, Sayaca C, Calik M, Doral MN. Effects on Lower Extremity Neuromuscular Control Exercises on Knee Proprioception, Muscle Strength, and Functional Level in Patients with ACL Reconstruction. *BioMed research international*. 2019:1-7.
4. Shen M, Che S, Ye D, Li Y, Lin F, Zhang Y. Effects of backward walking on knee proprioception after ACL reconstruction. *Physiotherapy theory and practice*. 2019.
5. Baltaci G, Harput G, Haksever B, Ulusoy B, Ozer H. Comparison between Nintendo Wii Fit and conventional rehabilitation on functional performance outcomes after hamstring anterior cruciate ligament reconstruction: prospective, randomized, controlled, double-blind clinical trial. *Knee surgery, sports traumatology, arthroscopy : official journal of the ESSKA*. 2013;21(4):880-887.
6. Beynon BD, Johnson RJ, Naud S, et al. Accelerated versus nonaccelerated rehabilitation after anterior cruciate ligament reconstruction: A prospective, randomized, double-blind investigation evaluating knee joint laxity using roentgen, stereophotogrammetric analysis. *American Journal of Sports Medicine*. 2011;39(12):2536-2548.
7. Fu CLA, Yung SHP, Law KYB, et al. The effect of early whole-body vibration therapy on neuromuscular control after anterior cruciate ligament reconstruction: a randomized controlled trial. *American journal of sports medicine*. 2013;41(4):804-814.
8. Risberg MA, Holm I, Myklebust G, Engebretsen L. Neuromuscular training versus strength training during first 6 months after anterior cruciate ligament reconstruction: a randomized clinical trial. *Phys Ther*. 2007;87(6):737-750.
9. Zult T, Gokeler A, van Raay J, et al. Cross-education does not accelerate the rehabilitation of neuromuscular functions after ACL reconstruction: a randomized controlled clinical trial. *European journal of applied physiology*. 2018;118(8):1609-1623.
10. Duval S, Tweedie R. Trim and fill: A simple funnel-plot-based method of testing and adjusting for publication bias in meta-analysis. *Biometrics*. 2000;56(2):455-463.
11. Ioannidis JP, Trikalinos TA. The appropriateness of asymmetry tests for publication bias in meta-analyses: a large survey. *CMAJ*. 2007;176(8):1091-1096.
12. Sanders TL, Maradit Kremers H, Bryan AJ, et al. Incidence of anterior cruciate ligament tears and reconstruction: a 21-year population-based study. *Am J Sports Med*. 2016;44(6):1502-1507.
13. Guyatt GH, Oxman AD, Vist GE, et al. GRADE: an emerging consensus on rating quality of evidence and strength of recommendations. 2008;336(7650):924-926.

VERSION 2 – REVIEW

REVIEWER	Seixas, Adérito Fundação Ensino e Cultura Fernando Pessoa
REVIEW RETURNED	13-Apr-2021
GENERAL COMMENTS	I would like to congratulate the authors for their effort improving the manuscript.
REVIEWER	Moore, Joe University of Portsmouth

REVIEW RETURNED	14-Apr-2021
GENERAL COMMENTS	Thank you for your clear responses to my comments. Overall I am happy that the changes you have made have sufficiently addressed my comments. Where you have chosen not to make changes I also appreciate these are more subjective comments and am therefore happy that your responses are appropriate.